# Assessment of Coating Quality Obtained on Flame-Retardant Fabrics by a Magnetron Sputtering Method

**DOI:** 10.3390/ma14061348

**Published:** 2021-03-11

**Authors:** Pamela Miśkiewicz, Magdalena Tokarska, Iwona Frydrych, Marcin Makówka

**Affiliations:** 1Faculty of Material Technologies and Textile Design, Institute of Architecture of Textiles, Lodz University of Technology, 116 Zeromskiego St., 90-924 Lodz, Poland; pamela.miskiewicz@dokt.p.lodz.pl (P.M.); iwona.frydrych@p.lodz.pl (I.F.); 2Faculty of Mechanical Engineering, Institute of Materials Science and Engineering, Lodz University of Technology, 1/15 Stefanowskiego St., 90-924 Lodz, Poland; marcin.makowka@p.lodz.pl

**Keywords:** flame-retardant fabric, magnetron sputtering, surface properties, EDS analysis, DigiEye system

## Abstract

Innovative textile materials can be obtained by depositing different coatings. To improve the thermal properties of textiles, aluminum and zirconium (IV) oxides were deposited on the Nomex^®^ fabric, basalt fabric, and cotton fabric with flame-retardant finishing using the magnetron sputtering method. An assessment of coating quality was conducted. Evenly coated fabric ensures that there are no places on the sample surface where the values of thermal parameters such as resistance to contact heat and radiant heat deviate significantly from the specified ones. Energy-dispersive spectroscopy was used for the analysis of modified fabric surfaces. Non-contact digital color imaging system DigiEye was also used. The criterion allowing one to compare surfaces and find which surface is more evenly coated was proposed. The best fabrics from the point of view of coating quality were basalt and cotton fabrics coated with aluminum as well as basalt fabric coated with zirconia. The probability of occurrence of places on the indicated sample surfaces where the values of thermal parameters (i.e., resistance to contact heat and radiant heat) deviated significantly from the specified ones was smaller for Nomex^®^ and cotton fabrics coated with zirconia and Nomex^®^ fabric coated with aluminum.

## 1. Introduction

To produce increasingly innovative textile materials with special properties, scientists are increasingly depositing different coatings on starting materials using physical or chemical vapor deposition processes [1,2,3,4]. Many different physical phenomena are involved in the PVD (physical vapor deposition) technique, which occurs at a pressure of 10^−5^–10 Pa. PVD methods use pure metals and gases as source materials instead of the harmful compounds found in CVD (chemical vapor deposition) methods. The costs of the deposition process are high, and the surfaces of coated elements need special treatment before the deposition. However, the process is characterized by high efficiency when using sputtering and good and very good physical and mechanical properties of coatings, while the technological process is considered ecological. That is why many scientists use selected PVD methods to modify the surface of textile materials [5,6,7,8,9,10,11].

Magnetron sputtering (MS) is a method of PVD that is used successfully for depositing metals on textile substrates. The magnetron sputtering method is based on a glow discharge realized at reduced pressure and in the presence of two mutually perpendicular fields: electric and magnetic [1,2]. The emitted electrons move along a helical path in a direction perpendicular to the directions of both fields along the magnetic field force lines. There is a crossing of fields, which leads to the closing of electrons in the excitation zone, which causes the frequency of ionizing collisions of the atmospheric gas to increase significantly. That phenomenon has a great influence on the intensification of plasma excitation and significantly increases the plasma density volume adjacent to the magnetron. A strong electric field accelerates positively charged ions towards the cathode, which, when hitting the cathode surface, causes intensive sputtering. Magnetron sputtering can be carried in an inert atmosphere. As a result, metallic coatings are produced. Magnetron sputtering can be also carried in an atmosphere that is a mixture of reactive gases, which results in the production of coatings that are compounds (e.g., metals with nitrogen, oxygen, or carbon).

The MS was used to produce thin metal coatings (aluminum, titanium and copper) and oxide coatings (aluminum oxide and titanium oxide) on non-woven and woven substrates [6,8]. Cotton, polypropylene, and needle-punched polyester nonwovens were used for the tests [8]. Thin aluminum coatings allowed for a combination of high electrical conductivity and high mirror reflection. Additionally, it was found that the magnetron sputtering method gives new possibilities in the area of production of advanced textile materials. The MS method was used to obtain metal electrically conductive transmission lines on specific textile substrates [9]. Spun-bonded polypropylene nonwovens and polyamide needled nonwovens were selected for research. The deposited material was copper. To obtain a smooth surface, the nonwoven fabric was melted using pressing with a polyolefin film at 120 °C for 120 s. It was observed that the longer the deposition process, the greater the decrease in the surface resistance. Conductive copper coatings with a surface resistance of 0.2 Ω were obtained on a spun-bonded polypropylene nonwoven fabric. According to the authors, the transmission lines obtained could be used in the protective clothing dedicated to emergency services. The PVD method enables new textile composites to be made for textronic applications [7]. Another example is the use of the pulsed magnetron sputtering method to produce a silver coating on polyester fabrics [10]. Tests results showed that the coated fabric retained the original mechanical properties that it exhibited before the surface modification. The coating obtained using the pulsed magnetron sputtering method showed better antimicrobial effectiveness compared to the reference fabric without coating. In research on shielding properties, a cathodic sputtering method was used to obtain other metallic coatings [11]. Copper was sprayed on the surfaces of polyester, polypropylene, and viscose nonwovens. The nonwovens had low surface weights because they were to be used as wallpaper. The obtained samples were characterized by good shielding efficiency. The best value of this parameter was obtained for the polypropylene nonwoven with a deposited copper coating. The important condition for obtaining a good quality material is to create the most continuous metal coating on the nonwoven’s surface.

In the area related to thermal properties, the MS method is also used for depositing coatings on the surfaces of basalt fabrics [5,12,13,14]. The modified fabric, when it meets the relevant requirements, is used as an element of the clothes protecting against high temperatures and hot factors. Chromium, aluminum, and zirconia (IV) oxide coatings of various thicknesses were deposited on the basalt fabric surfaces. The modified fabric was subjected to a test of resistance to contact heat and resistance to radiant heat. Resistance to contact heat was carried out for contact temperatures of 100 °C and 250 °C; however, no coated fabric achieved the 2nd efficiency level of protection. With an increase in the thickness of the coating, as in the case of the 18 µm thick ZrO_2_ coating, the 1st efficiency level of protection for contact heat resistance was achieved for a contact temperature of 100 °C [12,13]. For metal coatings, no increase in the contact heat resistance was observed with increased coating thickness. In the case of the radiation heat resistance test, better results were obtained for fabrics with metal coatings (chrome, aluminum), due to the silver color of the modified sample [5]. The conducted research allowed the authors to state that the direct application of coatings to basalt fabric slightly improved the tested protective properties. Therefore, further work concerning the production of basalt composite is needed. Preliminary research shows that the composite exhibits significant improvement in the case of resistance to contact heat and radiant heat [14].

Obtained coatings are used to make new materials or to improve the thermal, mechanical, electrical, chemical, and biological properties of existing textiles. The coatings produced present significant differences in their characteristics [3]. The differences are in terms of structure, surface morphology (roughness, porosity), and mechanical properties, and they require an assessment, depending on the final product application. To assess the quality and thickness of a deposited metal coating or surface roughness, scanning electron microscopy (SEM) is used [11,15,16]. SEM with an energy-dispersive spectroscopy (EDS) system is a research method used in material engineering research primarily [1,17]. A defined area of the sample surface is subjected to a concentrated and focused electron beam that exhibits specific energy. The original electron beam penetrates deep into the surface layer of material and then induces in it various signals that come from the tested layer. The excited and analyzed electron signal allows imaging of the observed surface. Excited and analyzed characteristic X-ray radiation allows for the determining of the elemental composition of the surface layer of the tested material or object. The depth at which the signal is induced is related to the depth of penetration of the electron beam in the material under examination and depends on the type of material and the energy of the primary beam of the electron beam. The EDS analysis is performed for polyacrylic coatings applied to polyester fabrics [18].

Analysis of uniformity coating applied to the fabric surface is conducted using the digital color imaging system DigiEye [5,19,20]. Non-contact digital color imaging system DigiEye enables colorimetric measurements from samples with an ultra-small area, featuring irregular surfaces according to the International Commission on Illumination (CIE) standard [21]. The total color difference ∆*E* is calculated between two areas using chosen CIE illuminant. The illuminant D65 is most often used. The illuminant is a statistical representation of average daylight with a correlated color temperature of approximately 6500 K. The total color difference is measured on a scale from 0 to 100. If ∆*E* < 1, the difference is not perceptible by human eyes. Two colors can be optically distinguished if ∆*E* ≥ 1. For ∆*E* equal to 1 or 2 the difference is perceptible through close observation. The difference is perceptible at a glance for ∆*E* greater than or equal to 3. 

The main aim of the study is to assess the quality of coating deposited on the basalt fabric, Nomex^®^ fabric, and cotton fabric, with the flame-retardant finishing, using the magnetron sputtering method. A coating of 5 µm thickness was deposited on the surface of selected fabrics, due to a statistical analysis showing that the assumed thickness is preferred [22]. The evenly coated fabric ensures that there are no places on the sample surface where the values of thermal parameters such as the resistance to contact heat and resistance to radiant heat deviate significantly from the specified ones. The criterion allowing for the comparison of surfaces and indication of which surface is more evenly coated is proposed.

## 2. Materials 

The fabrics selected for testing are used in personal protective equipment (i.e., protective clothing or gloves used for protection against heat and flames).

Three flame-retardant twill weave fabrics were selected for testing [22]:Basalt fabric (100% basalt fibers) with a thickness of 0.55 mm and surface mass of 398 g/m^2^;Nomex^®^ fabric (93% m-aramid, 5% p-aramid, and 2% antistatic fibers P140) with a thickness of 0.37 mm and surface mass of 266 g/m^2^; andCotton fabric with the flame-retardant finish (100% cotton fibers) and a thickness of 0.66 mm and surface mass of 376 g/m^2^.

Basalt fibers are flame-retardant. Fabrics made of basalt fibers can be used as materials to protect against fire and heat. Nomex^®^ fibers are used in protective fabrics, garments, and insulation. These materials are considered to be flame-retardant textiles. Cotton flame-retardant fabrics can protect workers from heat and flames and have a soft hand-feel. In preliminary tests, the thermal conductivity coefficient was determined as described in [5]. Measurements were repeated ten times, and the coefficient of variation did not exceed 2.5%. The thermal conductivity coefficient of basalt fabric is 0.038 Wm^−1^K^−1^, Nomex^®^ fabric is 0.043 Wm^−1^K^−1^, and cotton fabric is 0.046 Wm^−1^K^−1^. All the fabrics were characterized as having low thermal conductivity.

Using a scanning electron microscope, the topography of unmodified basalt fabric (B), Nomex^®^ fabric (N), and cotton fabric with the flame-retardant finishing (C) were recorded at the same total visual magnification of 50× and presented in Table 1.

The surfaces of the samples were modified with aluminum (Al) and zirconium (IV) oxide (ZrO_2_). The thermal conductivity coefficient of aluminum is 237 W/(mK) and zirconium (IV) oxide is 2 W/(mK) at 20 °C. Aluminum is an excellent reflector of radiant energy; therefore, it was expected that this coating would protect against radiant heat. Zirconium (IV) oxide has a matte surface. Unlike aluminum, ZrO_2_ is a dielectric. Therefore, zirconium (IV) oxide coated materials were expected to have a good resistance to contact heat. 

In total, nine samples were prepared for the tests. The following designation was used for unmodified and modified fabrics:B—unmodified basalt fabricBA—Basalt fabric modified with Al coatingBZ—Basalt fabric modified with ZrO_2_ coatingN—unmodified Nomex^®^ fabricNA—Nomex^®^ fabric modified with Al coatingNZ—Nomex^®^ fabric modified with ZrO_2_ coatingC—unmodified cotton fabric with a flame-retardant finishingCA—Cotton fabric with a flame-retardant finishing, modified with Al coatingCZ—Cotton fabric with a flame-retardant finishing, modified with ZrO_2_ coating

Aluminum or zirconium (IV) oxide coatings of 5 µm thickness were deposited on only one side of the selected fabrics. The Al and ZrO_2_ coatings were chosen because they improved the thermal properties of textile substrates in previous research [22].

## 3. Methods

### 3.1. Magnetron Sputtering Method

Al and ZrO_2_ coatings were deposited using magnetron sputtering in the B-90 vacuum chamber (Hoch-Vacuum, Dresden, Germany). Four independent planar magnetrons WK100 with medium-frequency power sources (Dora POWER SYSTEMS, Wroclaw, Poland) were used for coating depositions. Magnetrons were fitted with circular targets of Ø100 × 10 mm; two were made from pure Al (4N) and two from pure zirconium (3.5N) for the deposition of Al and ZrO_2_ coatings, respectively. Magnetrons were fixed facing each other in a horizontal plane and on the same axis. Textile substrates and pure p-type monocrystalline wafers of (111) orientation and dimensions 10 mm × 10 mm × 0.525 mm before mounting in the vacuum chamber were cleaned with isopropyl alcohol. After mounting the specimens in the specimens’ holder, the vacuum chamber was closed and depressurized. The process of deposition was carried out after achieving a residual pressure of 2 × 10^−3^ Pa. In the first step, a specimen holder was put in motion around the vertical axis at a rotation speed of 8 rpm. Pure Ar (5N) was introduced into the vacuum chamber with a flow of 25 sccm. When the pressure in the vacuum chamber was stabilized, appropriate magnetron power sources were switched on, and the deposition of pure Al coating or pure Zr interlayer occurred. For the deposition of ZrO_2_ coatings, pure O_2_ (5N) was introduced into the vacuum chamber after deposition of the pure Zr interlayer. It should be noted that the bias was not used during the processes of deposition, so there was no electric field in the volume between the samples and the magnetrons, which would significantly accelerate the ionized sputtered material and thus increase the energy of the material reaching the surface of specimens. Keeping a low energy for sputtered atoms and particles lowered the temperature deposition process, which prevented polymeric substrates from degrading at elevated temperatures. Detailed parameters of coatings deposition are shown in Table 2.

### 3.2. SEM-EDS Method

The condition of coatings on fibers and the chemical composition of deposited films were determined with the use of the JEOL JSM-6610 LV (Jeol, Tokyo, Japan) scanning electron microscope, working in a low vacuum mode with an attached X-MAX 80 module (Oxford Instruments, High Wycombe, UK) using energy-dispersive spectroscopy with the following parameters: accelerating voltage 20 kV, pressure 50 Pa, and beam current (spot size) of medium value, set for the best focus on the image. Maps of chemical composition were recorded using the same parameters and with a resolution of 1024 (number of pixels in the X dimension over which the beam scans), and pixel dwell time was equal to 100 µs.

The EDS analysis allowed one to estimate the chemical composition of coatings produced on flame-retardant fabrics selected for testing.

### 3.3. DigiEye System

The DigiEye system (VeriVide, Leicester, UK) was used in the assessment of coatings quality. For this purpose, the sample surface was divided into smaller, but even, areas (i.e., nine squares with a side length of 1 cm (Figure 1)) according to the concept presented in [5].

The following neighboring pairs of squares were taken into consideration: AB, DE, GH, BC, EF, HK, AD, BE, CF, DG, EH, and FK. Determination of total color difference Δ*E* for the pairs, described in detail in [5], enabled us to detect any differences on the whole coated surfaces. Based on the results, the minimum value (Δ*E_min_*), the maximum value (Δ*E_max_*), and the mean value (Δ*E_mean_*), with a standard deviation (*SD*) were determined for each sample’s surface coating.

It should be noted that a low standard deviation *SD* did not mean that the fabric sample was evenly coated. Comparable values of ∆*E* obtained for the fabric samples meant that the total color differences between pairs of areas were comparable. This can be referred to as a two-colored chessboard. It indicates that the sample surface is not evenly coated.

A certain criterion was proposed to assess the quality of the coating. The condition that must be met was as follows:∆*E_max_* < 3(1)

This condition means that there are no pairs of squares on the sample surface whose colors differ significantly. A total color difference less than or equal to 3 is perceived as a significant color deviation.

If the condition was met, the following quality indicator could be determined:*D* = ∆*E**_min_*/∆*E**_mean_*(2)

Values of indicator *D* are in the range of (0,1). If *D* = 1, then the sample surface is evenly coated. The criterion allows one to compare surfaces and find which surface is more evenly coated.

### 3.4. Thermal Properties

Two parameters were selected to describe the thermal properties of unmodified and modified fabric samples.

The resistance to contact heat was determined based on the standard ISO 12127-1:2015 [23]. The OTI device (OTI Greentech AG, Berlin, Germany) for testing the thermal insulation at a contact temperature of 100 °C was used. The threshold time *t*_100_ corresponding to the time from the moment of first contact with the heating cylinder until the temperature of the calorimeter increased by 10 °C from the initial value (100 °C) was measured. Based on the standard EN 407:2020 [24], an efficiency level of protection was obtained when the threshold time of contact of the sample with the heating cylinder, which was heated to the selected temperature, was greater than or equal to 15 s. If the condition was met for a contact temperature equal to 100 °C, then the 1st efficiency level of protection against contact heat was reached. The 2nd, 3rd, and 4th efficiency levels were obtained for 250, 350, and 500 °C, respectively.

The resistance to radiant heat was determined according to the standard ISO 6942:2002 [25]. The relative heat transfer index *RHTI*_24_ was determined based on standard EN 407:2020 [24]. The 1st efficiency level of protection against radiant heat was obtained when *RHTI*_24_ ≥ 7 s, the 2nd one when *RHTI*_24_ ≥ 20 s, and the 3rd one when *RHTI*_24_ ≥ 50 s.

## 4. Results and Discussion

Three different substrates were chosen for testing. As seen in Table 1, the image of basalt fabric is smooth and slippery. In the uncoated basalt fabric, fraying of the edges of the material was observed. The structure density of basalt fabric was lower than that of Nomex^®^ and cotton fabrics. This was due to the greater thickness of basalt yarns. In the cases of Nomex^®^ and cotton fabric, non-weave yarns were observed on the surface of fabrics. The surfaces were not uniform.

The chosen textile substrates were coated with aluminum or zirconium (IV) oxide with a thickness of 5 µm. An assessment of resistance to contact heat based on the threshold time *t*_100_ and resistance to radiant heat based on the relative heat transfer index *RHTI*_24_ was conducted [22] and presented in Table 3. Measurements were repeated three times. The coefficient of variation did not exceed 2.4% and 2.8% in the case of *t*_100_ and *RHTI*_24_, respectively.

Two colors (green and red) were used in Table 3 to distinguish the results of measurements carried out for modified fabrics. The green color indicated that a certain level was reached (i.e., the 1st efficiency level of protection against contact heat and the 2nd efficiency level of protection against radiant heat). The red color means that no threshold had been reached.

The assessment of coatings was conducted based on the analysis of the chemical composition of coatings and an analysis of total color differences on the modified sample surface. Analyses of the chemical composition of coatings deposited on the fabrics were conducted using the EDS method. The number of main elements and remaining elements in each coating was determined. The related surface composition of chemical elements is depicted in Table 4. Due to the limitations of the EDS method, and especially the inability to determine the content of light and heavy elements at the same time, the values were estimated to one decimal place.

In the case of basalt fabric, many more elements were distinguished during EDS analysis because the approximate chemical composition of basalt rock is known. The chemical composition of basalt fibers expressed in %wt. is as follows: 51.6–59.3 SiO_2_, 14.6–18.3 Al_2_O_3_, 5.9–9.4 CaO, 3.0–5.3 MgO, 9.0–14.0 FeO + Fe_2_O_3_, 0.8–2.3 TiO_2_, 0.8–2.3 Na_2_O + K_2_O, and 0.09–0.13 others [26]. Unfortunately, we did not know the content of the elements in Nomex^®^ and cotton fabrics with the flame-retardant finishing. In the case of aluminum coating, the thickness of the coating was sufficiently large that in chemical composition analysis there was no information from the substrate undercoating. One can only assume that the detected aluminum may have partly come from the substrate. For the other two fabrics, the coating was thinner, and during EDS analysis, the pieces of substrate and elements forming the fabric were visible. For a zirconia coating, the number of main elements (i.e., Zr and O) was comparable for all substrates. The presence of carbon in the chemical composition of aluminum and zirconia coatings resulted from the contamination of the sample surface. The appearance of this element in the case of SEM-EDS analysis is a well-known problem [27]. This problem can be solved by preliminary ion etching of the surface of the tested sample; however, in the case of the presented work this treatment was not applied and will be taken into consideration in future work.

Images of fabric surfaces taken on a scanning electron microscope at 200× magnification are presented in Table 5 and Table 6.

The main chemical constituents present in the aluminum and zirconia coatings are also presented in the form of color images in Table 5 and Table 6, where the color indicates the presence of a chemical component.

From an analysis of images, better adhesion of aluminum to all types of substrates was observed. The images of fabrics with the zirconia coating showed numerous cracks, which resulted from the fact that ceramics are fragile and their adhesion to the substrate is worse.

Using the DigiEye system, the total color differences between the two areas on the sample surface were determined for all modified fabric samples. The total color differences for the pairs of squares for each sample are presented in a graphical form and juxtaposed in Figure 2. Each value of the total color difference is placed on the border of defined two areas on the sample surface (see Figure 1).

Parameters describing color differences on samples were calculated and shown in Table 7. Additionally, the standard deviation *SD* was calculated.

Results indicate that there were some pairs of squares on sample surfaces where differences in colors could be optically distinguished (∆*E_max_* ≥ 1). A total color difference ∆*E* above 3, perceived as a significant color deviation, was not observed. Therefore, Equation (1) was met. Therefore, all fabric coatings were able to be compared based on values of quality indicator *D*, Equation (2). The order of the fabric samples with the most evenly coated surfaces are as follows: CA, BZ, BA, NZ, CZ, and NA. As shown in Figure 2, the first three sample (CA, BZ, BA) coatings were the best from the point of view of coating quality. In the case of the last three samples (NZ, CZ, NA) values of resistance to contact heat and radiant heat may have depended on the measuring place chosen on the sample surface. In the case of Nomex^®^ and cotton fabrics, non-weave yarns were observed on the surface of fabrics. Thus, the unmodified substrates already showed some unevenness.

## 5. Conclusions

Scanning electron microscopy (SEM) with an energy-dispersive spectroscopy (EDS) system and the DigiEye system are tools that enable the assessment of coating quality, especially thin coatings obtained on flame-retardant fabrics using a magnetron sputtering method.

In the case of aluminum coating, the coating was thick for basalt fabric (BA). For the other two fabrics (NA, CA) the coating was thinner, but the amount of the main element (Al) was comparable. For a zirconia coating, the number of main elements (i.e., Zr and O) were comparable for all textile substrates.

Many cracks were observed on the zirconia coating, resulting from the fragility of ceramics and its worse adhesion. From this point of view, the aluminum coating is better. The best fabrics from the point of view of coating quality are CA, BZ, and BA. Thus, the probability of occurrence of places on the sample surfaces where the values of thermal parameters, such as resistance to contact heat and resistance to radiant heat, deviate significantly from the specified ones is smaller than for NZ, CZ, and NA fabrics.

## Figures and Tables

**Figure 1 materials-14-01348-f001:**
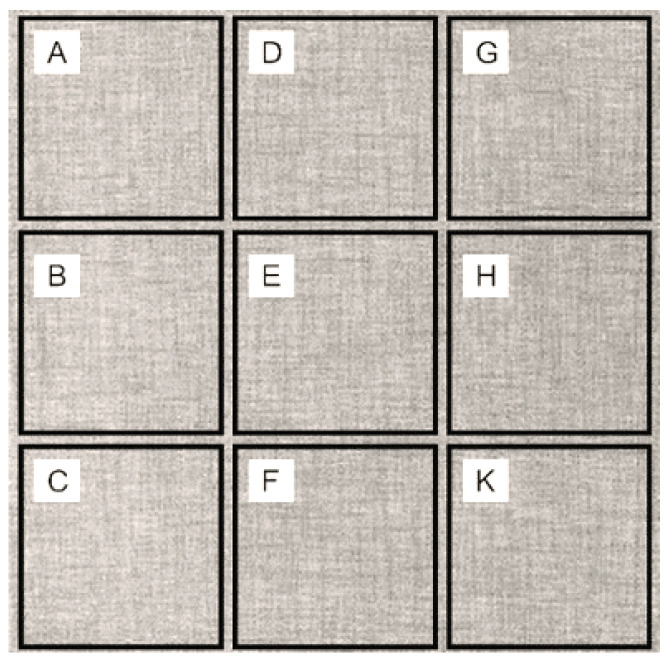
Sample surface division.

**Figure 2 materials-14-01348-f002:**
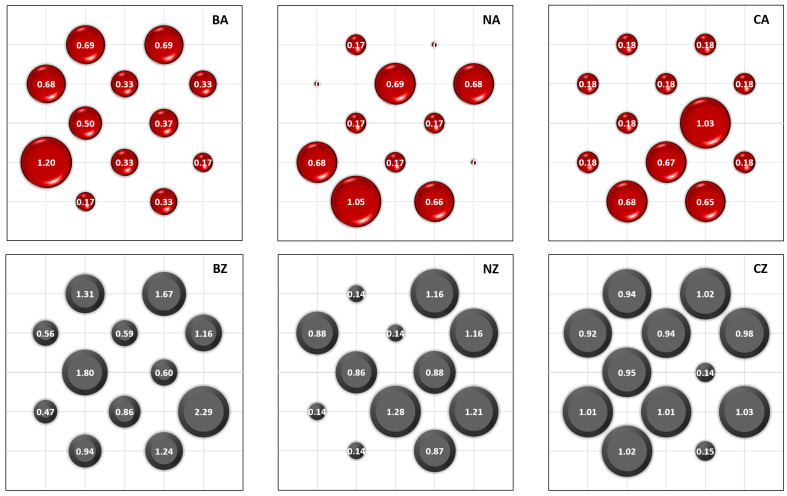
Total color differences for pairs of squares for each modified fabric.

**Table 1 materials-14-01348-t001:** SEM images of fabrics.

Basalt Fabric	Nomex^®^ Fabric	Cotton Fabric
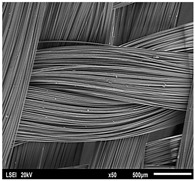	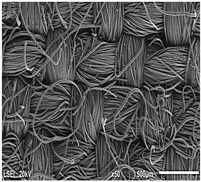	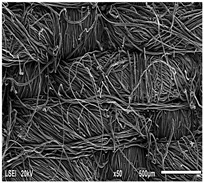

**Table 2 materials-14-01348-t002:** Parameters of the MS process.

Coating	Process Time (min)	Residual Pressure (Pa)	Process Pressure (Pa)	Flow of Ar and O_2_ (sccm)	Power on Magnetrons (kW)
Al	140	~2 × 10^−3^	3.7–3.8 × 10^−1^	25; 0	2 × 1.0 (Al)
ZrO_2_	175	~2 × 10^−3^	3.8–4.0 × 10^−1^	25; 12–13	2 × 1.5 (Zr)

**Table 3 materials-14-01348-t003:** Results of measurements of resistance to contact heat and resistance to radiant heat.

Sample	B	BA	BZ	N	NA	NZ	C	CA	CZ
*t*_100_, s	10.8	11.2	14.9	12.3	11.9	13.5	12.5	12.6	12.5
*RHTI*_24_, s	12.3	24.5	11.4	12.9	23.1	9.7	14.9	20.7	11.2

**Table 4 materials-14-01348-t004:** Chemical composition of fabrics coating.

Sample	Amount of the Main Elements (wt.%)	Amount of the Remaining Elements (wt.%)	Sample	Amount of the Main Elements (wt.%)	Amount of the Remaining Elements (wt.%)
BA	Al—99.4	Si—0.2 Fe—0.4	BZ	Zr—55.2 O—26.5	C—14.6 Si—1.5 Al—0.7 Fe—1.5
NA	Al—78.2	C—17.3 O—4.5	NZ	Zr—64.0 O—21.4	C—14.6
CA	Al—86.0	C—12.7 O—1.3	CZ	Zr—63.2 O—23.3	C—13.5

**Table 5 materials-14-01348-t005:** Analysis of main chemical constituents present in the aluminum coating.

Sample	BA	NA	CA
SEM image	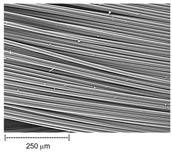	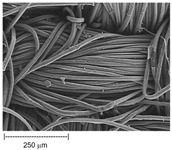	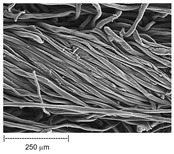
Aluminum presence	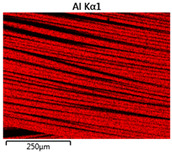	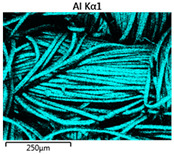	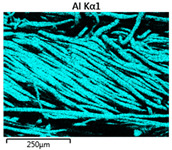

**Table 6 materials-14-01348-t006:** Analysis of main chemical constituents present in the zirconia coating.

Sample	BZ	NZ	CZ
SEM image	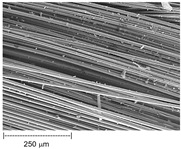	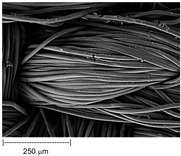	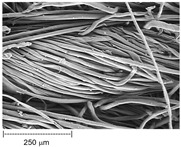
Zirconium presence	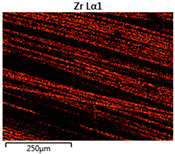	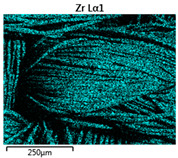	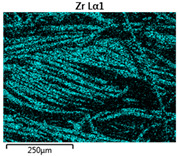
Oxygen presence	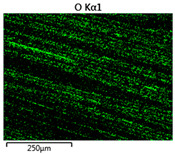	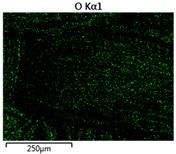	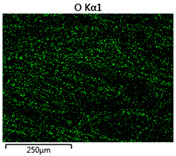

**Table 7 materials-14-01348-t007:** Parameters describing color differences on samples.

Sample	BA	BZ	NA	NZ	CA	CZ
Δ*E_min_*	0.17	0.51	0.01	0.14	0.18	0.14
Δ*E_max_*	1.20	2.29	1.05	1.28	1.03	1.03
Δ*E_mean_* (*SD*)	0.48 (0.29)	1.12 (0.57)	0.37 (0.36)	0.74 (0.46)	0.37 (0.30)	0.84 (0.33)
*D*	0.35	0.46	0.03	0.19	0.49	0.17

## Data Availability

Experimental methods and results are available from the authors.

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
