# Peer review of "Assessment of Coating Quality Obtained on Flame-Retardant Fabrics by a Magnetron Sputtering Method"

_materials, 2021, doi:10.3390/ma14061348_

Round 1
Reviewer 1 Report
The authors report on the realization and characterization of 3 different fabric with different coatings, aiming to optimize the heat resistence.
The experimental part is rather broad but I think that the paper is limited to a description of the results of the performed characterization, without any particular understanding of the origin of the results. This limits the contribution significance to the discussed samples and do not allow to obtain general conclusions for the fabric coating optimization.
Moreover in few points the authors describe quantitative criteria for the results classification, but it is not clear if the numeric limits are arbitrary or have some motivation. For example the colorimentric analysis, based on the difference of the R value (that is not defined), is based on the idea that Delta R above 1 results in color difference optically distinguishable and Delta R above 3 in "significant color variation". It there any particular meaning in the 1 and 3 thresholds?
In a similar way the thermal properties characterization is resumed in Figure 2 (that is just a sequence of number) in which 3 colors are present, but only green and red are defined in some way.
Overall the main limit of the paper is the lack of conclusions of general validity, that limits its interest to these specific fabrics, coated with specific layer in a given deposition parameters setting. The authors should deeply revise the data analysis on order to try to understand the origin of the different samples, without limiting to observe that they are different
Reviewer 2 Report
The present work is devoted to improving the properties of textiles by creating protective coatings on the fibre surface by magnetron sputtering. This is a very interesting and important technical problem. The topic of this work is fully consistent with "Materials". The results presented in the article are of interest to a wide range of readers of the journal. However, to publish an article, authors must take into account a number of recommendations.
- In the introduction section, authors should clearly explain their choice of coating materials.
- Lines with numbers 138-150; 170-182; 194-200, which describes the principles of methods of obtaining and investigating coatings, are redundant in the section "Materials and Methods". I think it's better to move them to the "Introduction" section.
- Section 3.4 should be shortened because most of the work was done in accordance with ISO standards.
- A feature of the magnetron sputtering method is the ability to obtain coatings on fibres with a melting point of up to 60 deg. [https://doi.org/10.1063/1.5017580]. I think the authors should point out this avenue in the Introduction section.
- The figures from Table 1 must be combined with the figures from Tables 4 and 5 and transferred to the Results section. In this case, the figures are easier to understand. One can clearly see how the treatment affects the properties of the fibres.
- The chemical composition of raw materials should be reported in Table 3.
- There is no statistics section in the work. The number of samples tested is not known for the data presented in the table (figure 2).
Round 2
Reviewer 1 Report
The authors revised the manuscript aiming to address my comments. Even if the performed revisions are limited to slight revision of the text the manuscript is improved. I thus suggest its acceptance in the present form.